# Distribution and Prevalence of *Anaplasmataceae*, *Rickettsiaceae* and *Coxiellaceae* in African Ticks: A Systematic Review and Meta-Analysis

**DOI:** 10.3390/microorganisms11030714

**Published:** 2023-03-09

**Authors:** Carlo Andrea Cossu, Nicola E. Collins, Marinda C. Oosthuizen, Maria Luisa Menandro, Raksha Vasantrai Bhoora, Ilse Vorster, Rudi Cassini, Hein Stoltsz, Melvyn Quan, Henriette van Heerden

**Affiliations:** 1Department of Veterinary Tropical Diseases, Faculty of Veterinary Science, University of Pretoria, Onderstepoort 0110, South Africa; 2Department of Animal Medicine, Production and Health, Faculty of Veterinary Medicine, University of Padova, Legnaro, 35020 Padova, Italy

**Keywords:** *Anaplasma*, *Rickettsia*, *Coxiella*, *Ehrlichia*, tick-borne disease, Africa

## Abstract

In Africa, ticks continue to be a major hindrance to the improvement of the livestock industry due to tick-borne pathogens that include *Anaplasma*, *Ehrlichia, Rickettsia* and *Coxiella* species. A systemic review and meta-analysis were conducted here and highlighted the distribution and prevalence of these tick-borne pathogens in African ticks. Relevant publications were searched in five electronic databases and selected using inclusion/exclusion criteria, resulting in 138 and 78 papers included in the qualitative and quantitative analysis, respectively. Most of the studies focused on *Rickettsia africae* (38 studies), followed by *Ehrlichia ruminantium* (27 studies), *Coxiella burnetii* (20 studies) and *Anaplasma marginale* (17 studies). A meta-analysis of proportions was performed using the random-effects model. The highest prevalence was obtained for *Rickettsia* spp. (18.39%; 95% CI: 14.23–22.85%), *R. africae* (13.47%; 95% CI: 2.76–28.69%), *R. conorii* (11.28%; 95% CI: 1.77–25.89%), *A. marginale* (12.75%; 95% CI: 4.06–24.35%), *E. ruminantium* (6.37%; 95% CI: 3.97–9.16%) and *E. canis* (4.3%; 95% CI: 0.04–12.66%). The prevalence of *C. burnetii* was low (0%; 95% CI: 0–0.25%), with higher prevalence for *Coxiella* spp. (27.02%; 95% CI: 10.83–46.03%) and *Coxiella*-like endosymbionts (70.47%; 95% CI: 27–99.82%). The effect of the tick genera, tick species, country and other variables were identified and highlighted the epidemiology of *Rhipicephalus* ticks in the heartwater; affinity of each *Rickettsia* species for different tick genera; dominant distribution of *A. marginale*, *R. africae* and *Coxiella*-like endosymbionts in ticks and a low distribution of *C. burnetii* in African hard ticks.

## 1. Introduction

Ticks are parasitic arachnids (phylum Arthropoda, class Arachnida) that feed only on the blood of vertebrate animals, including mammals, birds, reptiles, and amphibians. Currently, there are three recognized tick families: *Ixodidae* (“hard ticks”), *Argasidae* (“soft ticks”), and *Nuttalliedae* (only one species) [1]. Generally, ticks harbor a wide variety of microbes, including endosymbionts, commensals and tick-borne pathogens (TBPs), that represent a complex microbiome [2,3,4]. Ticks (specifically, *Ixodidae*) are regarded as the second major vectors—after mosquitos—that transmit pathogens to humans and animals [5], and many TBPs can coexist simultaneously within the same tick vectors, having either synergistic or antagonistic interactions [6,7,8]. Currently, TBPs constitute causative agents of the world’s most serious emerging infectious diseases, and in Africa, ticks continue to be a major impediment to the improvement of the livestock industry [9]. *Anaplasmataceae* and *Rickettsiaceae* (order Rickettsiales) are two families of obligate intracellular bacteria that parasitize eukaryotes. Currently, the family *Anaplasmataceae* includes five genera (*Anaplasma*, *Ehrlichia*, *Neoehrlichia*, *Neorickettsia*, and *Wolbachia*) [10]. Since the discovery of the human pathogens, *Ehrlichia chaffeensis*, which causes human monocytotropic ehrlichiosis (HME), and *Anaplasma phagocytophilum*, which causes human granulocytic anaplasmosis (HGA), in the 1980s and 1990s, the incidence of diseases caused by *Anaplasma* and *Ehrlichia* spp. has steadily increased in both developed and developing countries [11,12,13]. Diseases of veterinary importance were regularly reported in ruminants, including *E. ruminantium* (heartwater), *A. marginale*, *A. centrale* and *A. bovis* (bovine anaplasmosis), while *A. platys* (canine cyclic trombocytopenia) and *E. canis* (canine monocytic ehrlichiosis) were detected in dogs [14,15].

The *Rickettsiaceae* family is made up of three genera, namely *Rickettsia, Orientia* and *Candidatus* Cryptoprodotis [16]. Based on different genetic, epidemiological and pathological features, pathogenic rickettsiae are classified in four lineages: typhus group (TG), spotted fever group (SFG); ancestral group (AG); and the transitional group (TRG), with TG and TRG being transmitted by mites, louse and fleas, and SFG and AG being transmitted by ticks. The most common zoonotic bacteria reported in Africa are the SFG rickettsiae, mainly represented by *Rickettsia africae*, *R. aeschlimannii*, *R. conorii* and *R.massiliae* [17]. African tick bite fever (caused by *R. africae*) is regarded as the second most frequent febrile illness reported in travelers returning from sub-Saharan Africa (SSA), with the incidence of rickettsial infections being as high as 5.6% [18,19]. The importance of rickettsial pathogens transmitted by ticks is increasing dramatically and novel Rickettsia species are continuously being detected, raising questions about their pathogenicity. Additionally, several species, previously classified as non-pathogenic, are now associated with human infections [20].

The bacterial family *Coxiellaceae* was historically included in the order Rickettsiales together with the *Anaplasmataceae* and *Rickettsiaceae*, but the analysis of 16S and 23S rRNA gene sequences led to its re-classification into the order Legionellales [10]. The family *Coxiellaceae* is composed of the genera *Coxiella, Rickettsiella* and *Aquicella* [21]. *Coxiella burnetii* is the most significant representative of this taxonomic group as it causes Q fever, an emerging disease with high impact on public health, animal health and the economy. Infection with *C. burnetii* is acquired by the inhalation of desiccated aerosol particles. Ticks are not considered essential for the transmission of *C. burnetii* in livestock, but they may play a role in the maintenance of the life cycle in wildlife [22,23]. Indeed, there is a possibility that *C. burnetii* replicates in the midgut of ticks and appears in the feces nine days after a blood meal [24], and that transmission through ticks could be associated with contaminated dust from dried tick excrement [25]. Nevertheless, there is no evidence for transmission to humans by ticks. *Coxiella*-like endosymbionts (CLEs) are a large group of yet-to-isolate and characterize bacteria, phylogenetically close to *C. burnetii*, often associated with ixodid ticks worldwide i.a. [26]. DNA barcoding using 16S rRNA gene sequence data identified a number of CLEs in ticks that were genetically distinct from *C. burnetti* [27]. Within the *Coxiella* genus, the 16S rRNA gene sequences from CLEs showed between 91–98% nucleotide identity, indicating the occurrence of genetic diversity within the genus [28]. CLEs are classified into four clades: clade A includes *C. burnetii* and CLEs of *Ornithodoros* ticks; clade B contains CLEs of *Haemaphysalis* ticks (e.g., *Haemaphysalis longicornis*, *Haemaphysalis obesa*) and a *Coxiella* sp. (H-JJ-10) that causes horse infection [29]; clade C has CLEs of *Rhipicephalus* ticks (e.g., *Rhipicephalus turanicus*, *Rhipicephalus sanguineus*) and strains that cause opportunistic human skin infections [30]; clade D includes small-genome CLEs of *Amblyomma* ticks (e.g., *Amblyomma americanum*, *Amblyomma cajennense*). According to Duron et al., 2015 [27], all strains of *C. burnetiid* are the descendants of a *Coxiella*-like progenitor. Contrary to this hypothesis, Brenner et al., 2021 [31] demonstrated that a common virulent ancestor gave rise to the clade A (*C. burnetiid* and CLEs sequenced from *Ornithodoros* ticks). Several other tick endosymbionts likely evolved from pathogenic ancestors, indicating that pathogen-to-endosymbiont transformation is widespread across ticks. Virulence genes have not been found in CLEs, but the main biosynthesis pathways of vitamins and cofactors are encoded in most CLEs. As a consequence, it is thought that CLEs might be involved in a mutualistic interaction with the tick host by compensating nutritional vitamin deficiencies, thus explaining the classification of the pathogen as an endosymbiont [27,32,33,34]. Like the CLEs, other endosymbionts (i.e., intracellular bacteria with a high prevalence and load that are generally transovarially transmitted) have been proven to be fundamental in the survival of ticks, including *Francisella*-like endosymbionts (FLEs) (order Thiotrichales), ‘*Candidatus* Midichloria’, *Wolbachia* and *Rickettsia* (order Rickettsiales) [3]. Although *C. burnetii* is considered the only pathogen within the genus *Coxiella*, other *Coxiellaceae* pathogens have also been identified, such as *Candidatus* Coxiella cheraxi, a pathogen of crayfish [35], *Candidatus* Coxiella avium, a pathogen of birds [36], and *Candidatus* Coxiella massiliensis, recently identified as a new agent of human infections causing atypical scalp eschar and neck lymphadenopathy syndrome, with a delayed evolution to crust eschar in the area of the tick bite [30,37]. Finally, a multiorgan infection with a *Coxiella*-like organism was regarded as the cause of death of a female eclectus parrot (*Eclectus roratus*) [38]. However, the significance of CLE infections in terms of public and animal health is still to be investigated and clarified. With this systematic review and meta-analysis, we aim to comprehensively merge qualitative and quantitative (prevalence) data from the fragmented epidemiological literature on *Anaplasmataceae*, *Rickettsiaceae* and *Coxiellaceae* in African ticks. In achieving our aim, we implemented an unbiased, original, automated and direct methodology to scope and model evidence-based epidemiological information, essential for planning future research (e.g., to estimate sample size, compare results, plan further studies, etc.), highlighting hotspots for microbial activity, and thus providing reliable tools for health authorities and decision-makers. Three main objectives were therefore set: (1) record and map the distribution of *Anaplasmataceae*, *Rickettsiaceae* and *Coxiellaceae* in African countries; (2) estimate pooled prevalence of selected pathogens in tick populations using meta-analysis; and (3) assess the statistical significance and impact of the determinants associated with pooled prevalence, using subgroup-analyses and meta-regression.

## 2. Materials and Methods

### 2.1. Search Strategy

This systematic review and meta-analysis are registered in the international database of prospectively registered systematic reviews (PROSPERO) with the following ID: CRD42022339139. To ensure this review has all the elements and characteristics required for a systematic review, the PRISMA checklist and an additional comprehensive checklist were provided by Migliavaca et al., 2020 [39] (see Appendix A). We used the PICO (Population Intervention Comparison Outcome) model to establish the research questions, search strategy and the inclusion/exclusion criteria. In particular, the population (P) of interest was ticks living in Africa; intervention (I) included laboratory detection tests, i.e., nucleic acid (molecular) tests, antigen tests or direct identification (e.g., microscopy); comparison (C) was the difference among tests of the same test group, e.g., polymerase chain reaction (PCR) vs real-time PCR; the outcome (O) of interests was the presence or absence of *Anaplasmataceae*, *Rickettsiaceae* and/or *Coxiellaceae*. Consequently, our research questions were: What laboratory tests are able to detect *Anaplasma*, *Rickettsia* and/or *Coxiella* in African ticks? Which of the target pathogens species have been detected in African ticks? What is the prevalence of the target pathogens in African ticks? What is the role, if any, of the target population in pathogen/disease epidemiology? To retrieve such information, we formulated the following search algorithm: “Africa AND tick AND (anaplasma OR ehrlichia OR rickettsia OR Coxiella)”. The algorithm was run in four different electronic databases: ScienceDirect, PubMed, Scopus and Ovid. In PubMed, MeSH terms were searched and entered in the search strategy in order to retrieve relevant publications (PubMed algorithm: Africa[MeSH] AND tick[MeSH] AND (anaplas-ma[MeSH] OR ehrlichia[MeSH] OR rickettsia[MeSH] OR Coxiella[MeSH])). An additional database, i.e., OAIster, was used to search for grey literature. Records were imported into the Mendeley Desktop (version 1.19.8), where duplicates were removed and the selection process completed.

### 2.2. Selection Process

Articles retrieved with our search strategy were initially screened by title and abstract, and subsequently a full-text examination. Articles were excluded according to one or more of the following exclusion criteria: (i) article type not applicable, i.e., poster session, interview, abstracts, symposia, oral presentations, review; (ii) study area not applicable, i.e., the study was not conducted in Africa; (iii) target population is not ticks; (iv) intervention not applicable, e.g., intervention was therapy and not diagnostics; (v) ticks were explicitly stated as engorged, because pathogens may be detected in the blood meal rather than in the tick itself; (vi) outcome not applicable, i.e., pathogens or microbes investigated differed from the target pathogens. A detailed list of the reasons why studies were excluded during the full-text examination is reported in Appendix A. While examining included manuscripts full-text, we retrieved one study that escaped the search strategy and we added it to our analyses. 

For meta-analysis, the following inclusion criteria were selected: (i) studies focused on hard ticks rather than soft ticks; (ii) studies using suitable quantitative molecular tests (no sequencing data; see “Qualitative and quantitative analyses” paragraph); (iii) data only obtained from analysis of individual ticks, i.e., results obtained from tick pools were excluded due to indirectness (see Results section, Qualitative analysis paragraph).

### 2.3. Data Extraction and Critical Assessment of Included Studies

Data were extracted for a total of 26 variables grouped into five categories: publication specifics, tick specifics, sample specifics, laboratory specifics and epidemiological specifics. Raw data were then entered and shared with all authors in a Google Sheet spreadsheet (Google sheet: systematic review on *Anaplasmataceae*, *Rickettsiaceae* and *Coxiellaceae* in African ticks). Concurrently with data extraction, a critical assessment of the risk of bias of individual studies was performed using a modified version of the Appraisal tool for Cross-Sectional Studies (AXIS). This appraisal tool consists of a checklist that includes 20 questions to be answered either as “yes”, “no” or “don’t know”. Questions regarding non-responders (i.e., questions number 7, 13, 14) were not considered in this study, as they were not applicable for non-human subjects. The risk of bias of papers with less than 50% positive answers was assessed as “high”, 50–70% positive answers as “moderate” and more than 70% positive answers as “low”.

### 2.4. Qualitative and Quantitative Analyses

Raw data were handled in the R studio software (version 2022.12.0+353), where a qualitative analysis was initially performed using descriptive statistics. The frequency distribution of different variables was either aggregated in summary tables or visualized using barplots and maps. Meta-analysis was conducted to estimate the pooled molecular prevalence for each pathogen investigated in African ticks. Molecular prevalence was interpreted as the probability that a member of the target population tests positive for a pre-established pathogen, using a molecular detection test (e.g., *Rickettsia* spp. or *A. marginale*) at a certain point in time. Following our interpretation, DNA sequencing served as confirmation of positive results obtained with molecular screening tools, but did not report the proportion of cases actually tested. As a consequence, sequencing was not considered a suitable molecular test for estimating the pooled prevalence between studies and was only included in the qualitative analysis. The components of our meta-analytic method are listed in the supplementary checklist [39] in Appendix A. Justification for the choice of each component is as follows: ○Random effects model: the objective of our meta-analysis was to estimate the mean of the distribution of the true prevalence of *Anaplasmataceae*, *Rickettsiaceae* and *Coxiellaceae* in African tick populations, discarding the assumption that there is one true effect size which is shared between all the included studies (belonging to the fixed effects model). This choice was made on the assumption that microbial prevalence may differ greatly among tick populations based on several variables.○Sidik–Jonkman variance estimator, with Hartung–Knapp adjustment: to retrieve more conservative results than the common DerSimonian–Laird method, indicated by wider confidence intervals (CI) [40].○Clopper–Pearson confidence interval for individual studies: as above, to obtain wider confidence intervals especially when sample size is small [41], hesnce to retrieve more conservative results.○Freeman–Tukey double-arcsine transformation: to avoid overestimation of the weight of studies reporting prevalence close to 0% or 100%. The final pooled estimate and 95% CIs were back-transformed to a proportion.○Higgins and Thompson’s I^2^ statistic and prediction interval (PI): to assess between study heterogeneity. The I^2^ statistic is defined as the percentage of variability in the effect measure that is not caused by the sampling error. Low heterogeneity is represented by I^2^ = 25%, values of 50% indicate moderate heterogeneity, while substantial heterogeneity is represented by I^2^ ≥ 75%. Finally, the PI provides a range between which to expect the effects of future studies to fall based on present evidence [42].○Subgroup analyses and multiple meta-regression: to investigate the heterogeneity between studies. In subgroup analyses, we hypothesized that studies in our meta-analysis did not originate from one overall population. We instead assumed that they fell into different subgroups and that each subgroup had its own true overall effect. Our aim was to reject the null hypothesis that there is no difference in effect measured between the subgroups. For each of the results having a moderate to high heterogeneity (i.e., I^2^ > 70%), we conducted a subgroup analysis where moderators/subgroups were chosen in advance: tick genus, tick species, sampling country, sampling period (categorized in “Before 2002”, “2002–2011” or “2012–2022”), tick origin (domestic animals vs. wild animals vs. environment), tick identification method, sampling strategy, molecular method and risk of bias. Unlike subgroup analyses, in multiple meta-regression, we used more than one predictor to explain variation in effects. A step-wise regression method was adopted to select predictors based on a statistical criterion, i.e., all the moderators that tested significant with the subgroup analysis were first included in the multiple meta-regression model and then removed one by one based on the model fit indexes (residual I^2^ and R^2^). ○The small-study-effects method was used to evaluate the presence of publication bias: according to Egger et al., 1997 [43], we assumed that only small studies with a high prevalence are published. This method relies on the evaluation of funnel plot asymmetry, assessed either qualitatively (visual inspection of the funnel plot) or quantitatively, using the Egger’s regression test. For this test, a *p* < 0.05 was interpreted as the presence of significant asymmetry in the funnel plot. When this condition was satisfied, we used the Duval and Tweedie Trim and Fill Method to adjust for funnel plot asymmetry, selecting the estimator L0 for imputing missing studies [44]. 

Our meta-analysis results were visualized in summary tables and maps. Codes and functions utilized for meta-analysis can be retrieved from the first author’s GitHub website, using the URL: https://github.com/CarlVet/Scientific_papers/blob/main/Meta_analysis_codes, accessed on 15 February 2023.

### 2.5. Quality Assessment of the Body of Evidence

To ensure appropriate methodologic consistency, we evaluated the quality of evidence (QoE) for our pooled prevalence estimates using the GRADE (grading of recommendations assessment, development, and evaluation) guidelines [45]. This method rates the QoE as high, moderate, low, or very low, which reflects our certainty/confidence that the study outcomes are representative of the true effects. To decrease subjectivity and inconsistency, we implemented a quantitative automatized GRADE rating based on specific thresholds/criteria directly calculated from the extracted data. The rating workflow was as follows:Initial QoE was based on the study design. In our case, the effect of interest was the molecular prevalence of pathogens in tick populations, which could only be reported by observational studies (prevalence-reporting surveys or cross-sectional studies) [46]. Consequently, the study design did not impact the QoE of our prevalence estimates and the initial QoE was, therefore, set to the same score (3.33) for all the studies.Five domains could downgrade the initial QoE to up to 0.67 points each. They were interpreted in the following way:Risk of bias: individual studies were classified as high, moderate or low risk of bias, using the AXIS tool. The risk of bias of each prevalence estimate was calculated as a weighted average of the papers included in the respective meta-analysis. Finally, if the average risk of bias was determined to be high, we decreased the QoE by 0.67 points, 0.33 points for moderate risk, while for low bias risk, no points were reduced.Publication bias: the QoE was downgraded for publication bias if the Egger’s test indicated significant asymmetry in the funnel plot (*p* ≤ 0.05).Imprecision: downgraded (−0.67 points) if the 95% confidence intervals are wider than 20% (i.e., error level > 20%).Inconsistency: our interpretation of inconsistency relied on the heterogeneity that was not explained by the determinants investigated. Therefore, the QoE was downgraded for inconsistency (−0.67 points) if initial (before meta-regression) and residual (after meta-regression) heterogeneity indices (i.e., I^2^) were higher than 75%.Indirectness: among the different interpretations of indirectness provided by the GRADE guidelines, we only considered the indirectness for intervention. More specifically, if the variable “Molecular test” significantly affected the estimated pooled prevalence during subgroup analysis (i.e., *p*-value of the test for subgroup differences < 0.05), we downgraded the QoE because of indirectness (−0.67 points). Indeed, significantly different results obtained with different molecular tests were due to moderate to high differences in test sensitivity and specificity that may create a biased estimate.Three domains could upgrade the QoE: large-effect, dose-response gradient and if residual confounding would only decrease the magnitude of the effect [47]. We considered the large-effect domain applicable to our study. In particular, we upgraded the QoE when a large magnitude of effect was present on either side, i.e., if the lower bound of the CIs was higher than 10% (considering that at least 1 out of 10 ticks was infected) or if the upper bound was less than 1% (considering that less than 1 out of 100 ticks was infected).

If the final score fell within the interval of 0 to 1, we rated the QoE as “Very low +”, 1 to 2 as “Low ++”, 2 to 3 as “Moderate +++”, and 3 to 4 as “High ++++”.

### 2.6. Reliability

For reliability, each author was randomly assigned an equal subset of papers to verify the data extraction and to perform their own critical assessment of the studies included. Any discrepancies were discussed and resolved between the authors.

### 2.7. Literate Programming and Search Update

All the components of the manuscript (text, figures, tables, hyperlinks, citations) were built with R language and written as codes in a R markdown document [48]. The latter was finally rendered into Word format (using the “officedown” package), in order to allow the authors undertaking the revision process to track changes. This approach, namely “literate programming” [49], was based on the idea that a computer program should be documented in a manner that is understandable to humans, thus creating a single document that links textual data with programming or code and their outputs (plots, tables, maps, etc.). Any changes applied to the raw data (in the Google Sheet) were then automatically updated in the manuscript. This method ensured that bias was lowered considerably during data handling, processing and writing. Following the termination of the reliability process, the application of the literate programming automatically updated the data in the manuscript when the original search strategy per database was modified to articles published between 2021 and 2022.

### 2.8. Abbreviations

The present manuscript dealt with the scientific nomenclature of two different biological categories (bacteria and ticks), of which the genus names often start with the same initials (e.g., *Rickettsia* and *Rhipicephalus* or *Anaplasma* and *Amblyomma*). In order to avoid misunderstandings, we arbitrarily decided to abbreviate only bacterial names when repeated in the text—except when they start the sentence and in tables—while the scientific names of ticks were always kept in full.

## 3. Results

### 3.1. Qualitative Analysis

According to our search strategy and selection process, a total of 123 papers were originally included in the qualitative analysis and 73 in the quantitative analysis (Figure 1). Following the search update, an additional 15 studies were included in our database for the qualitative analysis and five in the quantitative analysis.

Most of the studies (95/136; 70%) included in our systematic review and meta-analysis were conducted in the last 10 years (2012–2022 period) (Figure 2), highlighting a substantial increase of interest and research in tick-borne bacteria.

The laboratory analyses were conducted mainly on individual tick samples (73% of the studies) (Figure 3). In other studies, ticks were pooled in several different ways or with an unclear or unexplained methods. On these premises, we decided to conduct the quantitative (meta-analytical) part of our study only on prevalence data obtained from individual tick samples.

A total of 21 species belonging to the family *Anaplasmataceae* were detected and identified in African ticks, the most represented being *E. ruminantium* (27 studies), followed by *A. marginale* (17), *A. platys* (12), *E. canis* (11), *A. phagocytophilum* (10) and *A. ovis* (9) (Table 1).

*Ehrlichia ruminantium* was reported across 13 sub-Saharan African countries (Figure 4) in a total of 14 tick species (*Amblyomma*: eight, *Rhipicephalus*: three and *Hyalomma*: three; Figure 5). In particular, *Amblyomma variegatum* (13 studies) has been found to be infected with *E. ruminantium* in nine African countries (Burkina Faso, Benin, Uganda, Ivory Coast, Cameroon, Gambia, Ethiopia, Sudan, Kenya), while *Amblyomma hebraeum* (10 studies) was reported to be infected with *E. ruminantium* in Southern African countries (South Africa, Swaziland, Zimbabwe) (Appendix A).

*Anaplasma marginale* has been detected in 17 tick species (11 *Rhipicephalus* spp., four *Amblyomma* spp., and one *Hyalomma* sp.; Figure 5) throughout Africa, except for the central part of the continent (Figure 4). In particular, *Rhipicephalus decoloratus* tested positive for infection with *A. marginale* in South Africa, Kenya, United Republic of Tanzania and Burkina Faso, while *A. marginale* was detected in *Amblyomma variegatum* in Benin, Madagascar and Ethiopia. *Anaplasma marginale* has a wide geographic distribution, as it is transmitted by several other tick species (Appendix A).

*Anaplasma platys* has been reported in 12 tick species (almost exclusively *Rhipicephalus* spp.; Figure 5) in seven African countries, i.e., South Africa, Kenya, Guinea, Ethiopia, Democratic Republic of the Congo, Tunisia and Egypt (Figure 4).

*Ehrlichia canis* has been reported in 10 countries (Figure 4). *Rhipicephalus sanguineus* (eight studies; Figure 5) was found to be infected with *E. canis* in six different countries in the northwestern part of the continent, while in the eastern part of the continent, the pathogen seems to be spread by other *Rhipicephalus* species (Appendix A). Unlike *E. ruminantium*, no African *Amblyomma* ticks have been found to be infected with *E. canis* thus far.

*Anaplasma phagocytophilum* has been reported in 14 tick species belonging to five different Ixodid tick genera (i.e., *Amblyomma*, *Hyalomma*, *Rhipicephalus*, *Haemaphysalis*, *Ixodes*) and two Argasid tick genera (i.e., *Argas* and *Ornithodoros*) (Figure 5). However, there was not one tick species in which *A. phagocytophilum* was most often detected, making *A. phagocytophilum* the most promiscuous *Anaplasma* pathogen in tick populations.

*Anaplasma ovis* was reported in a total of 11 tick species (mostly *Rhipicephalus* spp.) (Figure 5). The geographic distribution was extended to six African countries, where it was mainly spread by *Rhipicephalus turanicus*, *Rhipicephalus bursa* and *Rhipicephalus sanguineus* in the north of the Sahara, and by other tick species to the eastern and southeastern sub-Saharan African countries (Figure 4).

*Anaplasma bovis* was reported in a total of 10 tick species (mostly *Rhipicephalus* spp.) (Figure 5). This pathogen has been detected in only three African countries (i.e., South Africa, Kenya and Tunisia), where it is mainly spread by *Rhipicephalus evertsi*.

A total of 25 *Rickettsia* species were identified in African ticks. The most reported *Rickettsia* species was *R. africae* (38 studies), followed by *R. aeschlimanni* (24 studies), *R. massiliae* (19 studies) and *R. conorii* (12 studies) (Table 1). *Rickettsia africae* was reported in 26 African tick species (Figure 5), mainly *Amblyomma variegatum* (18 studies), across 17 African countries, i.e., in 71% of the total number of countries where *R. africae* has been reported (Figure 6). The main African countries that detected infection with *R. africae* in ticks were Kenya (nine studies), South Africa (four studies) and Ethiopia (four studies).

*Rickettsia aeschlimanni* was detected in 13 tick species (Figure 5), mostly *Hyalomma* (69% of total tick species) and mainly from *Hyalomma rufipes* (12 studies), *Hyalomma truncatum* (six studies), *Hyalomma impeltatum* (six studies) and *Hyalomma marginatum* (four studies). *Rickettsia massiliae* and *R. conorii* were detected almost exclusively in *Rhipicephalus* spp. and *Haemaphysalis* spp. (nine and six tick species, respectively), and mainly in *Rhipicephalus sanguineus* (nine and six studies, respectively; Figure 5). The northwestern part of the African continent has reported the highest prevalence of these *Rickettsia* species, although they have also been reported in central–southern African countries (Figure 6).

Regarding the *Coxiellaceae* family, *C. burnetii* (20 studies) was far more reported than CLEs and unidentified *Coxiella* spp. (four and five studies, respectively; Table 1). Nevertheless, the tick species and number are similar for all the reported *Coxiella* species (Figure 5). Indeed, numerous tick species belonging to the genera Ixodidae and Argasidae (~43) are known to be infected with the *Coxiella* species. *Coxiella* species have been reported in ticks throughout the continent, from north (Algeria, Tunisia, Egypt) to south (South Africa and Namibia), and from west (Senegal, Cote d’Ivoire, Nigeria, Sao Tome and Principe) to east (Ethiopia and Kenya). Epidemiological data, specifically from central Africa, are lacking (Figure 7).

To summarize, the studies focused on *Amblyomma* ticks reported mostly *Rickettsiaceae* and *Anaplasmataceae* infections (42/83 and 28/83 studies, respectively), especially *R. africae* (31/130 datasets), *Rickettsia* spp. (26/130 datasets), *E. ruminantium* (23/130 datasets) and *C. burnetii* (9/130 datasets). Additionally, most studies on *Hyalomma, Rhipicephalus* and *Haemaphysalis* ticks reported infection with *Rickettsiaceae* and *Anaplasmataceae* (Figure 8). Most infections detected in *Hyalomma* ticks were *Rickettsia aeschlimanni* (24/90 datasets), followed by *Rickettsia* spp. (15/90 datasets), *R. africae* (11/90 datasets) and *C. burnetii* (5/90 datasets); in *Rhipicephalus* ticks, mainly the *Rickettsia* spp. (23/166 datasets), followed by *R. massiliae* (17/166 datasets), *C. burnetii* (12/166 datasets) and *E. canis* (11/166 datasets); in *Haemaphysalis* ticks, mainly the *Rickettsia* spp. (11/33 datasets), *R. massiliae* (4/33 datasets) and *C. burnetii* (3/33 datasets). The remaining information on the other tick genera are reported in Figure 8. This figure should not be interpreted as tick–pathogen preferences, but rather a factor of number of investigations and positive reports.

### 3.2. Quantitative Analysis

Meta-analysis was performed for a total of 17 target bacteria, detected using molecular tests in African hard ticks (Table 2). The pooled prevalence of *Ehrlichia* and/or *Anaplasma* species in individual samples of African hard ticks, based on genus- and species-specific molecular techniques, was generally quite low (~0–1%), reaching 0% for *A. centrale*, *A. bovis* and *A. phagocytophilum*. The highest prevalence estimates were obtained for *A. marginale* (12.75%; 95% CI: 4.06–24.35%), *E. ruminantium* (6.37%; 95% CI: 3.97–9.16%) and *E. canis* (4.3%; 95% CI: 0.04–12.66%). The PI was the widest for *A. marginale* (0–84.73%), and narrower for *E. canis* (0–37.44%) and *E. ruminantium* (0–27.85%), indicating that range estimates of future prevalence are more accurate for the two latter pathogens. The results for these pathogens show considerable heterogeneity (I^2^ > 85%), making it justifiable to investigate the association with eventual determinants.

The pooled prevalence of the *Rickettsia* species in individual samples of African hard ticks, based on genus- and species-specific molecular techniques was generally higher than the *Anaplasmataceae* prevalence (i.e., ~3–18% vs. ~0–13%, respectively) (Table 2). The highest prevalence estimates were obtained for the *Rickettsia* spp. (18.39%; 95% CI: 14.23–22.85%), *R. africae* (13.47%; 95% CI: 2.76–28.69%) and *R. conorii* (11.28%; 95% CI: 1.77–25.89%). The PI was quite wide for all the *Rickettsia* species, meaning that we might find much higher prevalence in future investigations. The results for *Rickettsia* pathogens also showed considerable heterogeneity (I^2^ > 85%).

*Coxiella burnetii* was, without any doubt, the most studied *Coxiella* species in Africa, as it was investigated in 139 datasets and more than 6000 ticks. However, the prevalence of *C. burnetii* has been estimated to be as low as 0% (95% CI: 0–0.25%), as well as the PI, indicating that future prevalence will not exceed 15.8%. On the other hand, the pooled prevalence obtained for the *Coxiella* spp. (27.02%; 95% CI: 10.83–46.03%) and *Coxiella*-like endosymbionts (70.47%; 95% CI: 27–99.82%) were much higher than *C. burnetii* pooled prevalence, although the PI indicates that future results are very variable.

Subgroup analysis revealed that the determinants mostly associated with pathogen prevalence were: tick genus (12/12 pathogens), tick species (12/12 pathogens), sampling country (10/12 pathogens), risk of bias (7/12 pathogens) and molecular test (7/12) (Table 3).

The molecular prevalence estimates of the target pathogen species in different tick genera and species are summarized in Table 4 and Table 5, respectively. Quantitative distribution in different African countries is represented in Figure 9.

According to the subgroup analysis, we combined the significant variables in multiple meta-regression models. The best-fitting models (i.e., the ones accounting for the highest amount of heterogeneity) are represented in Table 6, column “formula”. The test of moderators was significant for most pathogens (except *Coxiella* spp. And *R. massiliae*), confirming that the selected variables do influence the prevalence of selected pathogens. The residual heterogeneity was still quite high (>75%) for numerous pathogens (*E. ruminantium*, *A. marginale*, *Rickettsia* spp., *R. africae*, *R. conorii*, *Coxiella* spp. and CLEs), meaning that the estimated prevalence differs also according to other variables not included in our study.

According to the Egger’s test, the estimates for *A. centrale*, *A. bovis*, *Rickettsia* spp. and *R. aeschlimanni* showed significant funnel plot asymmetry, thus indicating the presence of publication bias. The trim-and-fill method then filled missing studies to adjust for funnel plot asymmetry (Figure 10).

### 3.3. Quality of the Body of Evidence

According to our automatic GRADE rating process, the QoE for the prevalence estimates of *A. bovis*, *A. phagocytophilum* and *C. burnetii* were evaluated as high, providing confidence that the true effects are similar to the estimated effects, while the pooled effects for *E. ruminantium*, *A. marginale*, *A. ovis*, *R. africae*, *R. aeschlimanni and R. conorii* had a low QoE, hence the true prevalence might have been markedly different from the estimated prevalence. Finally, the prevalence of *Ehrlichia/Anaplasma* spp., *E. canis*, *A. centrale*, *A. platys*, *R.* spp., *R. massiliae*, *Coxiella* spp., and *Coxiella*-like endosymbionts resulted in a moderate QoE, indicating that the true effect was probably close to the estimated effect (Table 7).

## 4. Discussion

*Ehrlichia ruminantium* causes heartwater, a severe and economically important disease of cattle, sheep, goats and wild ruminants, limited to regions of SSA [25]. The pathogen is believed to be transmitted transstadially by several three-host ticks belonging to the genus *Amblyomma*, mainly *A. hebraeum* [9]. Nevertheless, our qualitative and quantitative analyses highlighted that tick species belonging to the genus *Rhipicephalus* may also be involved in the epidemiology of heartwater. Indeed, *E. ruminantium* was also reported in *Rhipicephalus microplus*, *Rhipicephalus decoloratus* and *Rhipicephalus sanguineus*, and with a high prevalence in *Rhipicephalus microplus*, since it was estimated at 14.21% [0–66.37%]. Such an unexpected result was justified by the authors [50] by a high rate of contact between *E. ruminantium* and *Rhipicephalus microplus* in western Africa due to high circulation of *E. ruminantium* [51,52] and a recent invasion of *R. microplus* in Benin [50]. However, other studies found that numerous *Rhipicephalus* ticks, tested in pools for the presence of *E. ruminantium*, were negative [53,54,55,56], raising the question as to whether such results are due to the low limit of detection and/or low parasitaemia, or whether they truly represent a negative outcome. The detection of *E. ruminantium* in the egg pool and progeny of (infected) *R. microplus* is concerning, but no experimental transmission of the pathogen by *R. microplus* to a susceptible host has been demonstrated.

*Anaplasma marginale*, together with *A. centrale*, is the agent of bovine anaplasmosis, known to be one of the most economically important diseases of the cattle industry on the African continent, especially in South Africa [57]. Infection of African ticks with *A. marginale* was reported in more than 15 studies (Table 1), in more than 15 tick species (Figure 5) from more than 10 African countries (Figure 4), and with a molecular prevalence higher than 10% (QoE = Moderate; Table 7). These numbers indicate that the risk of *A. marginale* transmission to cattle (the main vertebrate host) from African ticks, especially *Rhipicephalus microplus*, is quite high. Indeed, intrastadial and transstadial transmission of the pathogen has already been well documented in *Rhipicephalus microplus* [57]. Our results report that *Amblyomma variegatum* may also be significantly involved in the epidemiology of bovine anaplasmosis, as ticks infected with *A. marginale* were from three countries very distant from each other (Benin, Ethiopia and Madagascar) and with a large effect (prevalence ~ 20%). All the other *Anaplasma* species targeted with meta-analysis gave a very low prevalence (under 1%), with low heterogeneity indices (all I^2^ were below 65%, except *A. ovis*; Figure 4), possibly meaning that the role of ticks in maintaining these pathogens might be considered negligible. In particular, the risk of transmission of the zoonotic pathogen, *A. phagocytophilum* to humans in Africa should be regarded as low.

As highlighted in the qualitative analysis (Table 2), studies focused on *Rickettsiaceae* identified *R. africae*, the etiological agent of African tick bite fever, as the most reported pathogen, followed by *R. aeschlimannii*, *R. massiliae* and *R. conorii*. *Rickettsia africae* is transmitted both transstadially and transovarially by *Amblyomma* ticks [17,58], which readily bite humans. The prevalence estimate of *R. africae* in *Amblyomma* ticks is around 25%, which suggests an extreme fitness of this *Rickettsia* spp. as *Amblyomma* vectors. Considering the previous assumptions, and that the prevalence of *R. africae* in *A. variegatum* (a tick that occurs in areas with widely different climatic conditions) exceeds 70% (i.e., at least 7 of 10 *A. variegatum* ticks are positive for *R. africae*), *Amblyomma* should be considered the main maintenance host of the pathogen, and the risk of transmission to humans from these ticks should be regarded as high.

The analysis highlighted that *R. africae* mainly infects the *Amblyomma* species [55,59,60,61,62,63,64,65,66], while *R. aeschlimannii* is found in the *Hyalomma* species [55,64,67,68,69,70,71,72,73,74,75,76,77,78,79,80,81], and *R. massiliae* and *R. conorii*, in the *Rhipicephalus* species [82,83,84,85,86,87]. Furthermore, in the test for subgroup differences for these pathogen species, the “tick genus” variable was always statistically significant (Table 3). However, this assumption cannot be verified due to the lack of epidemiological (especially quantitative) data. Furthermore, even though qualitative and quantitative data suggest that tick species in the genera *Haemaphysalis*, *Dermacentor* and *Ixodes* may play a role in the epidemiology of the *Rickettsia* spp., not many studies, focused on reporting these infections in Africa, are available for these tick genera [60,73,85,88,89,90,91,92,93,94,95,96]. *Dermacentor* and *Ixodes* are considered the reservoirs of some SFG Rickettsias (e.g., *R. slovaca* and *R. helvetica*, *R. monacensis*, respectively) in other countries [97,98,99].

Although infection of ticks with *C. burnetii* was reported in more than 20 studies throughout Africa (Appendix A), the pooled molecular prevalence of this pathogen in individual ticks, collected in numerous African countries, was really low. Moreover, the QoE of this estimate was high, meaning that we are confident that the estimated prevalence is in fact very low. The test that was mainly used to detect the pathogen was real-time PCR, which gave almost exclusively negative results. These results corroborate that ticks are not efficient vectors for the maintenance and transmission of *C. burnetii*, but rather just act as sporadic mechanical vectors to vertebrate hosts [24,25]. However, significantly a higher prevalence was registered in *Ixodes* (3.3%; 95% CI: 1–6.6%) and *Heamaphysalis* (8.7%; 95% CI: 0–50.2%) ticks.

According to our prevalence estimates, the probability of detecting CLEs in African *Amblyomma*, *Dermacentor* and *Haemaphysalis* ticks cluster at around 100%, with the interval estimates indicating a lower limit of 30% (Table 5). These results show a remarkable fitness of CLEs for most African ticks. Since the pathogenicity of CLEs is still debated and epidemiological data are lacking from most countries (Figure 7), questions arise if they can constitute a major public health concern.

Based on the tick vector distribution [9] and estimated prevalence (Table 5), we may expect the presence of different pathogens in non-investigated countries: *E. ruminantium* in Botswana, Madagascar, Zambia, Tanzania, Democratic Republic of the Congo, Nigeria and Ghana, and *R. africae* in Madagascar, Zimbabwe, Botswana, Zambia, Tanzania, South Sudan, Camerun, Benin, Togo and Ghana through *Amblyomma hebraeum, Amblyomma variegatum* and/or *Rhipicephalus microplus*; *A. marginale* in Mozambique, Zimbabwe, Botswana, Uganda, Democratic Republic of the Congo, South Sudan, Sudan, Central African republic, Camerun, Nigeria, Togo and Ghana through *Amblyomma variegatum*; *R. aeschlimannii* in Namibia, Botswana, Zimbabwe, Mozambique, Zambia, Tanzania, Burundi, Uganda, Somalia, Eritrea, Benin, Togo, Ghana and Guinea through *Hyalomma truncatum*; and *R. massiliae* in all southern African countries through *Haemaphysalis leachi.* However, available data are not sufficient to make any risk assessment or prediction, which would require the collection and analysis of several different environmental, geographical and epidemiological variables, and the use of articulated spatial and ecological models.

A limitation of this study are that the meta-analysis was based only on results obtained from screening tests. As the pairwise nucleotide sequence homologies of SFG Rickettsia are >98.8% for the 16S rRNA gene, >92.7% *gltA* gene, >85.8% *ompB* gene and >82.2% gene D [100], screening techniques might have flawed our estimates due to lack of specificity of the molecular tests used. The occurrence of cross-reactions is to be considered also for the *Anaplasma* species, as they are most often detected by amplification (or amplification and sequencing) of small fragments of the 16S rRNA gene. The 16S rRNA sequences of many of the *Anaplasma* spp. are very similar, and if the full-length gene is not sequenced, it is not always possible to distinguish between the *Anaplasma* species. Therefore, it is possible that some authors misclassified certain *Anaplasma* species occurrences, as already highlighted by [101]. Another limitation is that some pathogens have only been investigated in a few countries or ticks, and their prevalence might vary markedly if searched elsewhere. In some instances, conventional PCR techniques detected more positives compared to real-time PCR, which is odd, as qPCR should be more sensitive. This finding leads us to doubt the specificity of some of the cPCR techniques used by the authors. 

Moreover, the pathogens might be detected in ticks because of indigested blood meal. When female adult ticks are collected from animals, they are almost always partially engorged, since they need up to 20 days to fully engorge with blood [9]. Although we limited this event by excluding studies that declared the use of engorged ticks, most of the publications did not specify the feeding status of tested ticks. As a consequence, the prevalence obtained from such data may be overestimated.

We did not include a number of determinants that may significantly affect our prevalence estimates or act as confounders, such as tick stage, tick sex, environmental variables (vegetation, soil, etc.) and climate variables (temperature, humidity, rainfall, etc.).

The trim-and-fill method indicated that the prevalence of the *Rickettsia* spp. and *R. aeschlimannii* might possibly be significantly smaller than we estimated because of publication bias, i.e., investigations having small sample sizes that obtained few or no positive results not being published.

The main observation from this study is a lack of standardization in determining the prevalence of TBP in African ticks. As highlighted in Figure 3, studies had several different tick pool sizes and strategies, and they were not always clear. Additionally, molecular methods used by the studies differed most often in technique and gene target. As a consequence, it was not possible to conduct a meta-analysis on pooled ticks because there was a significant indirectness of investigation. Furthermore, randomization and justification of the sample sizes were very rarely considered by the studies included in our work. Only 6/136 studies (4%) satisfied question no. 3 (regarding the justification of sample sizes) (Appendix A) of the AXIS tool, and only 20/136 of the studies (15%) satisfied question no. 6 (which indicated if randomization was present). We hereby suggest investigating TBPs in individual tick samples rather than pools, to provide quantitatively comparable results that can be added to the batch for statistical analysis. We also recommend, when possible, to apply randomization to the sampling strategy, thus providing more reliable results and a lower risk of selection bias.

## 5. Conclusions

With the present work, we comprehensively pooled all the epidemiological literature on *Anaplasmataceae*, *Rickettsiaceae* and *Coxiellaceae* in African ticks. We highlighted and discussed the main qualitative findings, and we provided reference values for the measure of prevalence. Moreover, we assessed the association and influence of several determinants for the prevalence of selected pathogens in African ticks. Considering the lack of standardization and data for the topic of interest throughout the African continent, this systematic review and meta-analysis can be used as a baseline for future epidemiological and/or experimental studies.

## Figures and Tables

**Figure 1 microorganisms-11-00714-f001:**
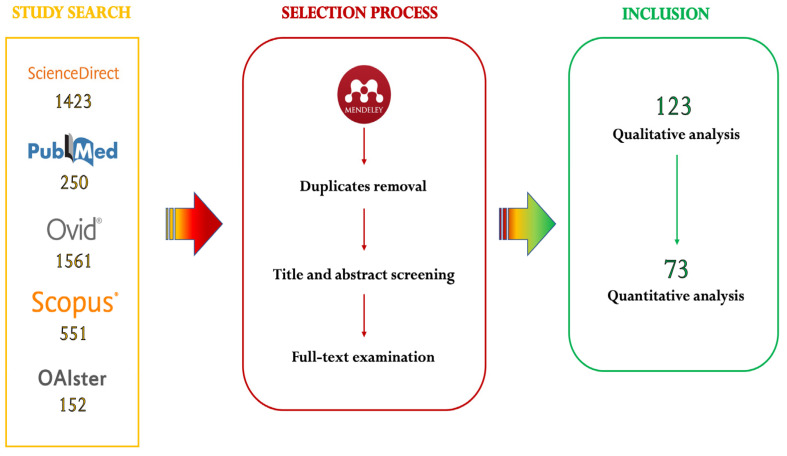
Papers included in our analyses according to our original search.

**Figure 2 microorganisms-11-00714-f002:**
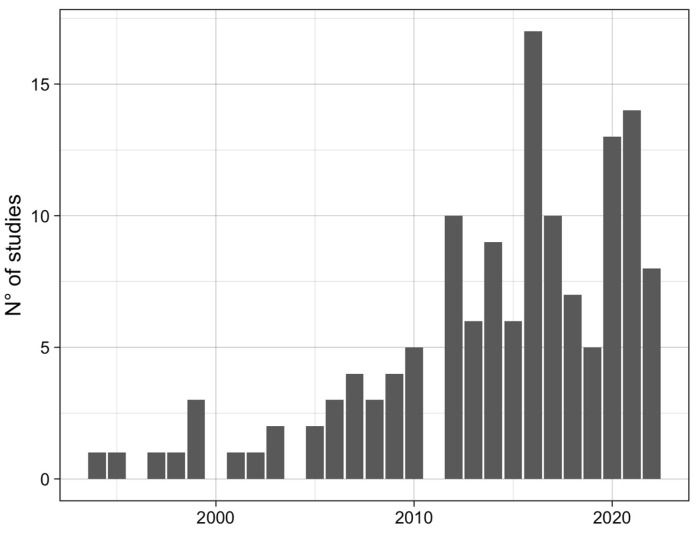
Number of studies on the detection of bacteria in ticks, belonging to the families *Anaplasmataceae*, *Rickettsiaceae* and *Coxiellaceae*, published from 1992–2022.

**Figure 3 microorganisms-11-00714-f003:**
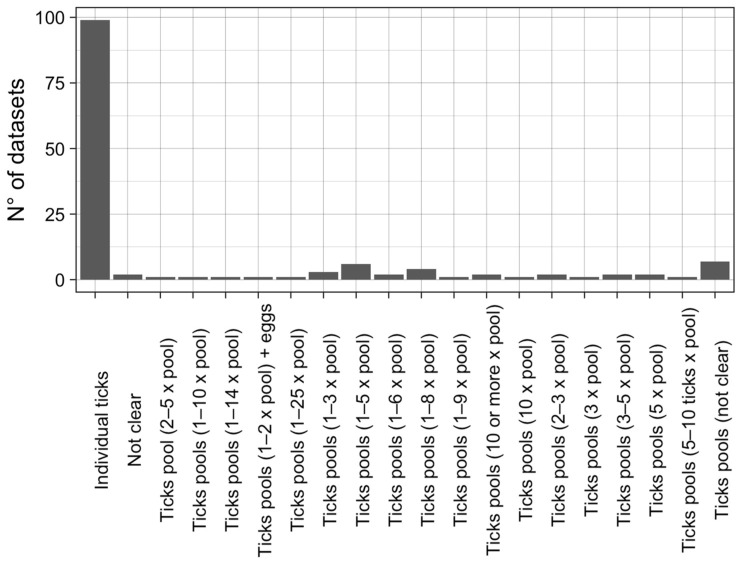
Number of datasets grouped per the variable sample type (individual ticks vs. pooled ticks).

**Figure 4 microorganisms-11-00714-f004:**
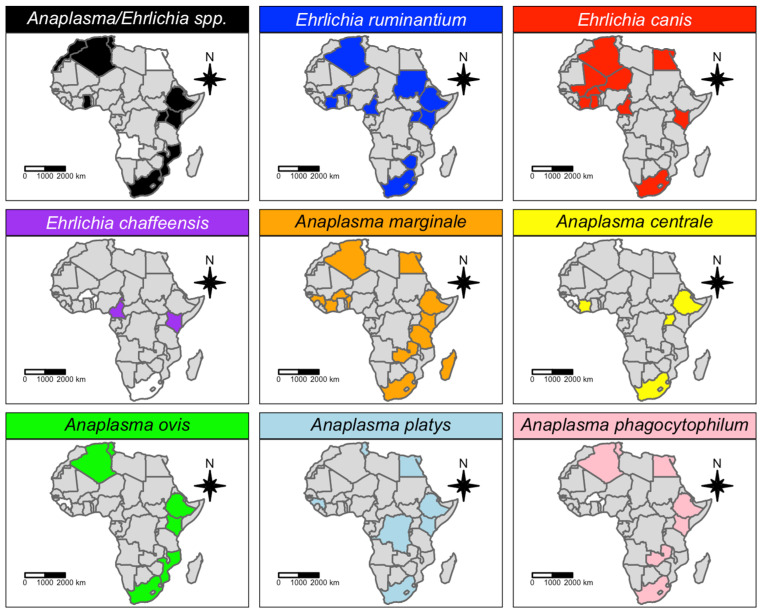
Geographic distribution of *Anaplasmataceae* species detected in African ticks. Countries where the pathogen was investigated, but not detected, are represented in white, while countries where the pathogen has not been investigated are represented in grey.

**Figure 5 microorganisms-11-00714-f005:**
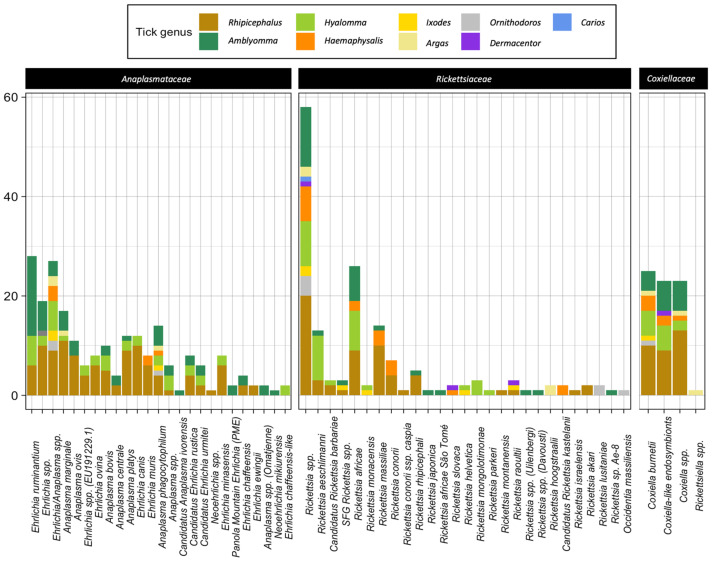
Number of tick species within each genus infected with different pathogen species.

**Figure 6 microorganisms-11-00714-f006:**
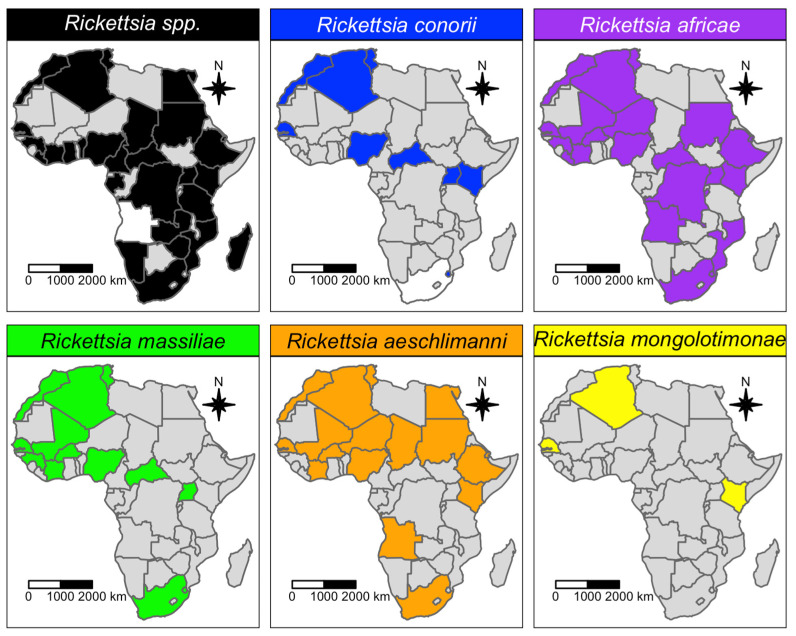
Geographic distribution of *Rickettsiaceae* species detected in African ticks. Countries where the pathogen was investigated, but not detected, are represented in white, while countries where the pathogen has not been investigated are represented in grey.

**Figure 7 microorganisms-11-00714-f007:**
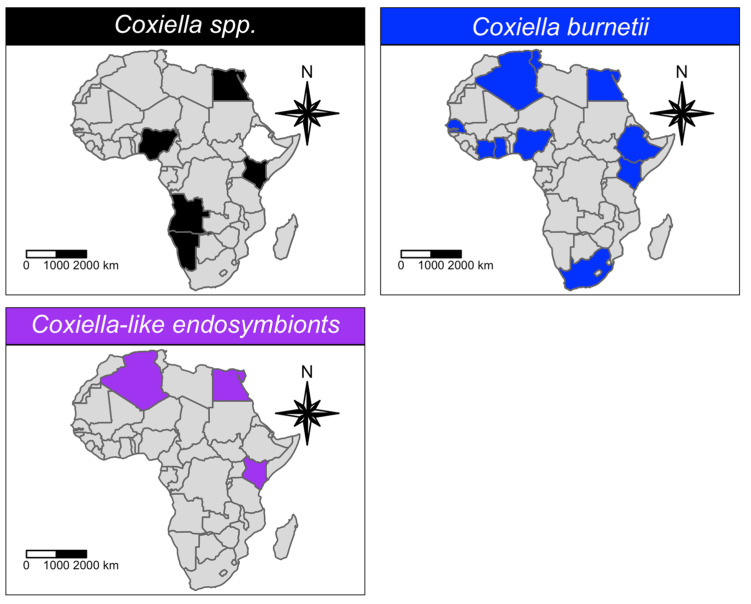
Geographic distribution of *Coxiellaceae* species detected in African ticks. Countries where the pathogen was investigated, but not detected, are represented in white, while countries where the pathogen has not been investigated are represented in grey.

**Figure 8 microorganisms-11-00714-f008:**
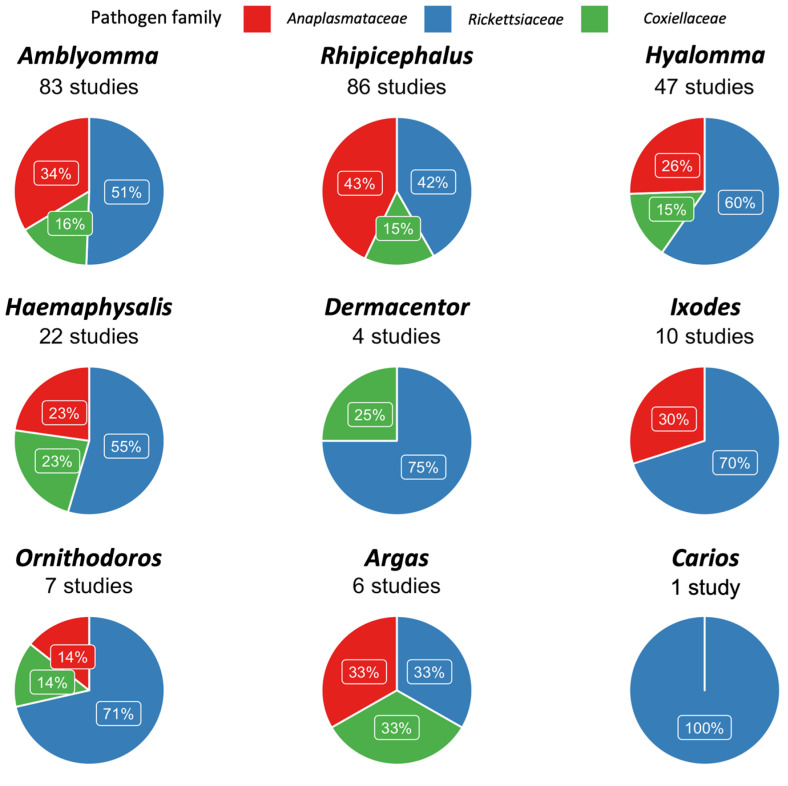
Relative percentage of the studies reporting infection with at least one *Anaplasmataceae, Rickettsiaceae* or *Coxiellaceae* species.

**Figure 9 microorganisms-11-00714-f009:**
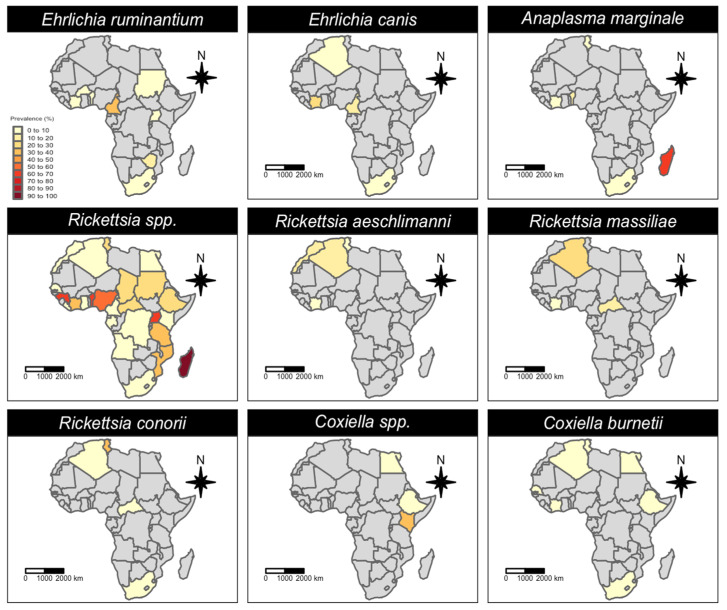
Choropleth maps showing the molecular prevalence of *Anaplasmataceae*, *Rickettsiaceae* and *Coxiellaceae* in African ticks. Only the estimates showing a significant association sampling country are here displayed. CLEs were reported only from two countries (Sao Tome and Principe and Algeria), hence their distribution is not represented here.

**Figure 10 microorganisms-11-00714-f010:**
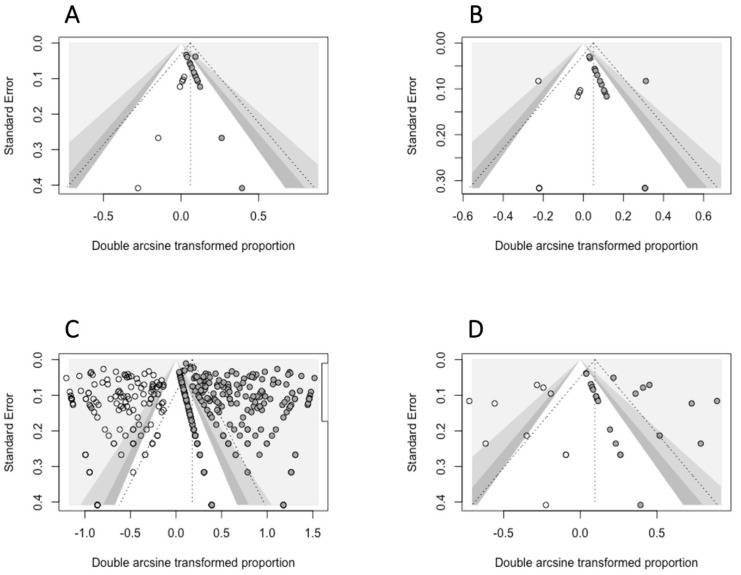
Contour-enhanced funnel plots of prevalence estimates that showed significant funnel plot asymmetry (Egger’s test *p* < 0.05). Solid-filled circles indicate the studies included in the original meta-analysis; empty circles indicate studies added by the trim-and-fill method to adjust for funnel plot asymmetry. (**A**) *A. centrale*; (**B**) *A. bovis*; (**C**) *Rickettsia* spp.; (**D**) *R. aeschlimannii*.

**Table 1 microorganisms-11-00714-t001:** Frequency distribution of *Anaplasmataceae*, *Rickettsiaceae* and *Coxiellaceae* pathogens detected in African ticks.

*Anaplasmataceae* Species	Studies	*Rickettsiaceae* Species	Studies	*Coxiellaceae* Species	Studies
*Ehrlichia ruminantium*	27	*Rickettsia* spp.	55	*Coxiella burnetii*	20
*Anaplasma marginale*	17	*Rickettsia africae*	38	*Coxiella* spp.	5
*Ehrlichia/Anaplasma* spp.	14	*Rickettsia aeschlimanni*	24	*Coxiella*-like endosymbionts	4
*Anaplasma platys*	12	*Rickettsia massiliae*	19	*Rickettsiella* spp.	1
*Ehrlichia canis*	11	*Rickettsia conorii*	12		
*Anaplasma phagocytophilum*	10	*Rickettsia monacensis*	8		
*Anaplasma ovis*	9	*Rickettsia helvetica*	4		
*Ehrlichia* spp.	7	*Rickettsia rhipicephali*	3		
*Anaplasma bovis*	5	*Rickettsia slovaca*	3		
*Anaplasma centrale*	4	*Rickettsia mongolotimonae*	3		
*Anaplasma* spp.	3	*Rickettsia raoultii*	3		
*Ehrlichia chaffeensis*	3	*Candidatus* Rickettsia barbariae	2		
*Ehrlichia muris*	2	*Rickettsia hoogstraalii*	2		
*Candidatus* Ehrlichia rustica	2	*Rickettsia lusitaniae*	2		
*Ehrlichia minasensis*	2	*Rickettsia conorii* ssp. caspia	1		
*Ehrlichia* spp. (EU191229.1)	1	*Rickettsia japonica*	1		
*Ehrlichia ovina*	1	*Rickettsia africae* São Tomé	1		
*Candidatus* Anaplasma ivorensis	1	*Rickettsia parkeri*	1		
*Candidatus* Ehrlichia urmitei	1	*Rickettsia montanensis*	1		
*Neoehrlichia* spp.	1	*Rickettsia* sp. (Uilenbergi)	1		
Panola Mountain *Ehrlichia* (PME)	1	*Rickettsia* sp. (Davousti)	1		
*Ehrlichia ewingii*	1	*Candidatus* Rickettsia kastelanii	1		
*Anaplasma* sp. (Omatjenne)	1	*Rickettsia israelensis*	1		
*Neoehrlichia mikurensis*	1	*Rickettsia akari*	1		
*Ehrlichia chaffeensis*-like	1	*Occidentia massiliensis*	1		

**Table 2 microorganisms-11-00714-t002:** Meta-analysis on the molecular prevalence of *Anaplasmataceae*, *Rickettsiaceae* and *Coxiellaceae* in African ticks.

Pathogen Species	N° of Datasets	N° of Ticks Tested	N° of Ticks Positive	Pooled Prevalence	95% CI (%)	95% PI (%)	I^2^ (%)
*Ehrlichia/Anaplasma* spp.	61	2295	184	**2.3%**	0.81–4.34	0–19.14	62.61
*Ehrlichia ruminantium*	44	7039	552	**6.4%**	3.97–9.16	0–27.85	89.82
*Ehrlichia canis*	9	508	47	**4.3%**	0.04–12.66	0–37.44	87.36
*Anaplasma marginale*	31	2322	455	**12.8%**	4.06–24.35	0–84.73	97.22
*Anaplasma centrale*	14	913	1	**0.0%**	0–0	0–0	0
*Anaplasma bovis*	14	879	3	**0.0%**	0–0	0–1.33	4.47
*Anaplasma ovis*	7	657	20	**0.6%**	0–3.73	0–10.99	81.34
*Anaplasma platys*	10	1271	22	**0.3%**	0–1.46	0–3.61	61.02
*Anaplasma phagocytophilum*	11	689	3	**0.0%**	0–0.15	0–0.74	0
*Rickettsia* spp.	326	14,188	3252	**18.4%**	14.23–22.85	0–95.75	96.63
*Rickettsia africae*	24	1391	285	**13.5%**	2.76–28.69	0–91.91	97.67
*Rickettsia aeschlimanni*	22	815	43	**2.6%**	0–9.48	0–45.03	83.55
*Rickettsia massiliae*	15	811	75	**6.9%**	0.21–18.43	0–59.89	92.06
*Rickettsia conorii*	16	679	77	**11.3%**	1.77–25.89	0–78.99	91.63
*Coxiella* spp.	32	341	97	**27.0%**	10.83–46.03	0–100	86.86
*Coxiella burnetii*	139	6442	493	**0.0%**	0–0.25	0–15.8	80.4
*Coxiella-like endosymbionts*	8	163	119	**70.5%**	27–99.82	0–100	94.81

The confidence interval (CI) indicates our 95% certainty that the true effect lies in the indicated range of values; the predictive interval (PI) provides a range between which to expect the effects of future studies to fall within.

**Table 3 microorganisms-11-00714-t003:** Statistical significance (*p*-values) of the moderators selected for our subgroup analysis.

	N° of Datasets	Tick Genus	Tick Species	Sampling Country	Sampling Period	Tick Origin	Tick Identification Method	Molecular Test	Risk of Bias
*Ehrlichia ruminantium*	44	**<0.001**	**0.042**	**0.001**	0.217	0.468	N/A	**<0.001**	**0.012**
*Ehrlichia canis*	9	**<0.001**	**<0.001**	**<0.001**	N/A	N/A	N/A	**<0.001**	**<0.001**
*Anaplasma marginale*	31	**0.001**	**<0.001**	**<0.001**	N/A	N/A	N/A	**<0.001**	**<0.001**
*Anaplasma ovis*	7	**0.003**	**<0.001**	N/A	N/A	N/A	N/A	N/A	**0.003**
*Rickettsia* spp.	326	**<0.001**	**<0.001**	**<0.001**	0.16	**0.02**	**<0.001**	**<0.001**	**0.038**
*Rickettsia africae*	24	**0.025**	**<0.001**	0.204	0.812	N/A	N/A	0.104	0.908
*Rickettsia aeschlimanni*	22	**<0.001**	**<0.001**	**0.042**	0.176	N/A	N/A	**<0.001**	**0.014**
*Rickettsia massiliae*	15	**<0.001**	**<0.001**	**<0.001**	**<0.001**	**<0.001**	N/A	0.241	0.156
*Rickettsia conorii*	16	**<0.001**	**<0.001**	**<0.001**	**<0.001**	0.127	N/A	**0.001**	0.051
*Coxiella* spp.	32	**<0.001**	**<0.001**	**<0.001**	1	0.122	N/A	N/A	1
*Coxiella burnetii*	139	**<0.001**	**<0.001**	**<0.001**	**<0.001**	0.796	0.22	0.494	**<0.001**
*Coxiella*-like endosymbionts	8	**<0.001**	**<0.001**	**<0.001**	N/A	N/A	**<0.001**	**<0.001**	N/A

**Table 4 microorganisms-11-00714-t004:** Estimated pooled prevalence in different tick genera.

Tick Genus	*E. ruminantium*	*E. canis*	*A. marginale*	*A. ovis*	*Rickettsia* spp.	*R. africae*	*R. aeschlimanni*	*R. massiliae*	*R. conorii*	*Coxiella* spp.	*C. burnetii*	CLEs
*Amblyomma*	**8%** [5.6–10.7%]	**0%** [0–0.1%]	**12.6%** [4.1–24.3%]		**56.6%** [45.7–67.2%]	**24.3%** [4.3–52.5%]	**0%** [0–1%]	**0%** [0–1%]	**0%** [0–0.1%]	**45.1%** [4.4–89.5%]	**0%** [0–2.1%]	**99.4%** [40.1–100%]
*Dermacentor*					**38.8%** [0.4–88.3%]					**0%** [0–38.9%]	**0%** [0–50%]	**100%** [30.3–100%]
*Haemaphysalis*					**12.2%** [0.3–32.3%]			**4.2%** [0–17%]		**27.3%** [4.4–57.9%]	**8.7%** [0–50.2%]	**100%** [30.3–100%]
*Hyalomma*	**0%** [0–0.4%]		**0%** [0–0.4%]	**0%** [0–0.1%]	**6.1%** [2.5–10.7%]	**13.9%** [0–100%]	**13.2%** [2.1–28.9%]	**0%** [0–0.4%]		**0%** [0–2.6%]	**0%** [0–1.2%]	**22.5%** [4.5–46.4%]
*Ixodes*					**5.9%** [0–27.4%]					**0%** [0–100%]	**3.3%** [1–6.6%]	
*Rhipicephalus*	**10.5%** [0–47.8%]	**11.6%** [1.7–27.2%]	**21.1%** [0–57.9%]	**2.7%** [0–10.4%]	**6.1%** [2.8–10.2%]	**1%** [0–5%]	**0%** [0–0%]	**14.9%** [2.8–32.4%]	**18.8%** [4.3–39%]	**37.4%** [12.4–65.6%]	**0%** [0–0.4%]	

**Table 5 microorganisms-11-00714-t005:** Estimated prevalence in different tick species.

Tick Species	*E. ruminantium*	*E. canis*	*A. marginale*	*A. ovis*	*R. africae*	*R. aeschlimanni*	*R. massiliae*	*R. conorii*	*C. burnetii*	CLEs
*Amblyomma astrion*									**0%** [0–4.1%]	**97.6%** [90.1–100%]
*Amblyomma cohaerens*									**5.1%** [0–31.4%]	
*Amblyomma gemma*									**24.4%** [13.1–37.1%]	
*Amblyomma hebraeum*	**9.4%** [5.3–14.3%]	**0%** [0–0.1%]	**0%** [0–0.1%]		**4.2%** [0–13.1%]			**0%** [0–0.1%]	**0%** [0–25%]	
*Amblyomma lepidum*	**1.9%** [0–5.6%]								**0%** [0–6.5%]	
*Amblyomma* spp.									**0%** [0–0%]	
*Amblyomma* *sylvaticum*									**0%** [0–4.2%]	
*Amblyomma* *variegatum*	**7.5%** [4.4–11.3%]		**19%** [7.7–33.5%]		**72.1%** [23–100%]	**0%** [0–1%]	**0%** [0–1%]		**0.3%** [0–5.9%]	**100%** [97.2–100%]
*Dermacentor* *marginatus*									**0%** [0–50%]	**100%** [30.3–100%]
*Haemaphysalis* *erinacei*									**46.9%** [29.7–64.4%]	
*Haemaphysalis leachi*							**4.2%** [0–17%]		**0%** [0–26.8%]	
*Haemaphysalis* *punctata*									**0%** [0–100%]	
*Haemaphysalis* *spinulosa*									**0%** [0–53.9%]	
*Haemaphysalis sulcata*										**100%** [30.3–100%]
*Hyalomma aegyptium*									**0%** [0–0.7%]	
*Hyalomma detritum*						**20%** [0–67.5%]				**25%** [0–79.3%]
*Hyalomma dromedarii*				**0%** [0–1.1%]		**14.2%** [0–79.3%]			**0%** [0–6.4%]	
*Hyalomma excavatum*				**0%** [0–18.3%]					**0%** [0–14.7%]	**36.4%** [17.3–57.8%]
*Hyalomma impeltatum*				**0%** [0–1.3%]		**0%** [0–44.4%]			**2.4%** [0–13.1%]	
*Hyalomma impressum*	**0%** [0–100%]		**0%** [0–100%]		**100%** [0–100%]	**0%** [0–88.8%]	**0%** [0–100%]		**0%** [0–100%]	
*Hyalomma lusitanicum*									**0%** [0–88.8%]	**33.3%** [0–94.1%]
*Hyalomma* *marginatum*	**0%** [0–10.5%]		**0%** [0–10.5%]		**12.5%** [0.3–34.1%]	**37.4%** [0–93.8%]	**0%** [0–10.5%]		**0%** [0–42.5%]	**14.3%** [3.3–30.1%]
*Hyalomma rufipes*									**3.1%** [0–15.2%]	
*Hyalomma scupense*									**0%** [0–99.3%]	
*Hyalomma truncatum*	**0%** [0–6.3%]		**0%** [0–6.3%]		**3.7%** [0–15.2%]	**11.1%** [1.5–26.3%]	**0%** [0–6.3%]		**2.1%** [0–14.3%]	
*Ixodes ricinus*									**0%** [0–0%]	
*Ixodes vespertilionis*									**15.8%** [2.3–36.2%]	
*Rhipicephalus* *annulatus*			**0%** [0–8%]	**0%** [0–8%]					**2%** [0–100%]	
*Rhipicephalus* *appendiculatus*					**1.1%** [0–4.6%]				**0%** [0–2.8%]	
*Rhipicephalus bursa*			**0%** [0–5.7%]	**0%** [0–5.7%]					**0.5%** [0–4.4%]	
*Rhipicephalus* *compositus*					**7.1%** [0–28.2%]					
*Rhipicephalus* *decoloratus*									**0%** [0–3.6%]	
*Rhipicephalus evertsi*									**0%** [0–0.4%]	
*Rhipicephalus guilhoni*									**0.5%** [0–8.3%]	
*Rhipicephalus* *lunulatus*							**4.3%** [0.1–12.7%]			
*Rhipicephalus* *microplus*	**14.2%** [0–66.4%]		**59.7%** [9.1–99.4%]		**3.2%** [0–16%]	**0%** [0–1.1%]	**0%** [0–1.1%]		**0%** [0–1.1%]	
*Rhipicephalus* *muhsamae*					**0.7%** [0–3%]		**6.9%** [3.3–11.7%]	**4.2%** [1.4–8.2%]	**0%** [0–6.1%]	
*Rhipicephalus* *praetextatus*									**0.8%** [0–4.1%]	
*Rhipicephalus* *pulchellus*									**21.3%** [7.5–38.7%]	
*Rhipicephalus* *sanguineus*		**11.6%** [1.7–27.2%]	**0%** [0–2.2%]	**2.5%** [0–7.5%]		**0%** [0–0%]	**25.1%** [11.2–41.9%]	**20.8%** [4.6–43.3%]	**0%** [0–1.5%]	
*Rhipicephalus* *senegalensis*	**0%** [0–50%]		**0%** [0–50%]		**0%** [0–50%]	**0%** [0–50%]	**60.1%** [0–100%]		**0%** [0–50%]	
*Rhipicephalus simus*									**0%** [0–100%]	
*Rhipicephalus* spp.									**0%** [0–5.6%]	
*Rhipicephalus sulcatus*							**3.1%** [0–12.9%]			
*Rhipicephalus* *turanicus*			**0%** [0–0.8%]	**7.9%** [4.7–11.8%]					**0%** [0–2.8%]	

Empty cells indicate non-investigated associations.

**Table 6 microorganisms-11-00714-t006:** Meta-regression on the molecular prevalence of *Anaplasmataceae*, *Rickettsiaceae* and *Coxiellaceae* in African ticks.

Pathogen Species	Formula	Residual Heterogeneity (I^2^),%	Amount of Heterogeneity Accounted for (R^2^),%	Test of Moderators (*p*-Value)
*Ehrlichia* *ruminantium*	Sampling_country * Test * Tick_species	78.4	67.03	**<0.001**
*Ehrlichia canis*	Sampling_country	50.58	80.32	**0.01**
*Anaplasma marginale*	Sampling_country + Tick_species	82.36	87.63	**<0.001**
*Rickettsia spp.*	Sampling_country * Tick_genus * Test * Risk_of_bias	87.86	57.85	**<0.001**
*Rickettsia africae*	Tick_species + Sampling_strategy	93.24	58.87	**0.007**
*Rickettsia* *aeschlimanni*	Tick_species + Test	55.16	74.28	**0.001**
*Rickettsia massiliae*	Tick_species + Sampling_country	59.78	77.68	0.07
*Rickettsia conorii*	Tick_species + Test	83.29	63.18	**0.003**
*Coxiella spp.*	Sampling_country * Tick_genus	83.63	18.58	0.158
*Coxiella burnetii*	Sampling country * Tick_species	55.04	35.8	**<0.001**
*Coxiella*-like *endosymbionts*	Sampling_country	83.32	50.22	**0.018**

**Table 7 microorganisms-11-00714-t007:** Quality of Evidence (QoE) for our prevalence estimates.

Pathogen Species	Pooled Estimate (%) [95% CI]	Reasons to Downgrade	Reasons to Upgrade	Score	Resulting QoE
*Ehrlichia*/*Anaplasma* spp.	2.32 [0.81–4.34]	Risk of bias ~ Moderate Indirectness		2.33/4	**Moderate +++**
*Ehrlichia ruminantium*	6.37 [3.97–9.16]	Risk of bias ~ Low Inconsistency Indirectness		1.99/4	**Low ++**
*Ehrlichia canis*	4.3 [0.04–12.66]	Risk of bias ~ Moderate Indirectness		2.33/4	**Moderate +++**
*Anaplasma marginale*	12.75 [4.06–24.35]	Risk of bias ~ Low Imprecision Inconsistency		1.99/4	**Low ++**
*Anaplasma centrale*	0 [0–0]	Risk of bias ~ Low, Publication bias Indirectness	Large effect	2.66/4	**Moderate +++**
*Anaplasma bovis*	0 [0–0]	Risk of bias ~ Low, Publication bias	Large effect	3.33/4	**High ++++**
*Anaplasma ovis*	0.55 [0–3.73]	Risk of bias ~ Low Inconsistency Indirectness		1.99/4	**Low ++**
*Anaplasma platys*	0.34 [0–1.46]	Risk of bias ~ Moderate		3/4	**Moderate +++**
*Anaplasma phagocytophilum*	0 [0–0.15]	Risk of bias ~ Low Indirectness	Large effect	3.33/4	**High ++++**
*Rickettsia* spp.	18.39 [14.23–22.85]	Risk of bias ~ Moderate Publication bias Inconsistency	Large effect	2.33/4	**Moderate +++**
*Rickettsia africae*	13.47 [2.76–28.69]	Risk of bias ~ Low Imprecision Inconsistency		1.99/4	**Low ++**
*Rickettsia aeschlimanni*	2.55 [0–9.48]	Risk of bias ~ Moderate Publication bias Indirectness		1.66/4	**Low ++**
*Rickettsia massiliae*	6.87 [0.21–18.43]	Risk of bias ~ Moderate		3/4	**Moderate +++**
*Rickettsia conorii*	11.28 [1.77–25.89]	Risk of bias ~ Moderate Imprecision Inconsistency		1.66/4	**Low ++**
*Coxiella* spp.	27.02 [10.83–46.03]	Risk of bias ~ Moderate Imprecision Inconsistency	Large effect	2.33/4	**Moderate +++**
*Coxiella burnetii*	0 [0–0.25]	Risk of bias ~ Low	Large effect	4/4	**High ++++**
*Coxiella*-like endosymbionts	70.47 [27–99.82]	Risk of bias ~ Low Imprecision Inconsistency	Large effect	2.66/4	**Moderate +++**

++++ is High; +++ is moderate; ++ is low; and + is very low QoE. These signs have always been used to indicate the QoE.

## Data Availability

Raw data are publicly available on Mendeley Data: https://data.mendeley.com/datasets/w7ghxty6tc/1, accessed on 15 February 2023.

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
