# Peer review of "Distribution and Prevalence of Anaplasmataceae, Rickettsiaceae and Coxiellaceae in African Ticks: A Systematic Review and Meta-Analysis"

_microorganisms, 2023, doi:10.3390/microorganisms11030714_

Round 1

Reviewer 1 Report

·       Title, and throughout the text – Anaplasmataceae, Rickettsiaceae, Coxiellaceae should not be in italic; reverse in Figure captions

·         Line 21 – prevalence was

·         Line 34 – feed only on the blood

·         Line 39 – vectors; that transmit; tick vector

·         Lines 41-42 – TBPs constitute causative agents of the world’s

·         Line 49 – diseases

·         Lines 51-52 – regularly were reported in ruminants including

·         Line 59 – TRG being transmitted by; AG being transmitted by

·         Line 76 – possibility

·         Line 77 – nine days

·         Line 120 – maybe is better health authorities, than managers

·         Line 133 – were provided by

·         Line 135 – are ticks

·         Line- 140 – questions were

·         Line 161 – tick itself; differed from the target pathogens

·       Lines 163-164 – initially excluded, and afterwards included back after full-text examination, or included additionally? Also in Line 319, please explain in more detail search update (inclusion of additional studies after initially selected)  

·         Line 174 – and shared

·         Line 189 – pre-established pathogen

·         Line 196 – start sentence in next row (for better overview of bullets)

·         Line 210 – between

·         Line 222 – in advance instead of preemptively

·         Line 224 – environment

·         Line 275 – pof? p of?

·         Lines 303-306 – 2012-2022?

·         Table 1 – in title: Anaplasmataceae, Rickettsiaceae; name of all species in one row; Neoehrlichia mikurensis  

·         Line 328 – Number of studies on the detection in ticks of bacteria belonging

·         Line 357 – Burkina Faso, while

·         Line 375 – and

·         Figure 5 – reorder families on figure to be the same as it is in the text itself: Anapl. Rick., Cox.

·         Figure 4 and 5 should be inversed, since Figure 5 regards to all of three families, while Figure 4 regards to Anaplasmataceae

·         Figure 8 – Carios 1 study

·         For all Tables – marks to be in one/more rows, but without word breaks (also, I guess that tables got some wrapping disruption during generating pdf)

·         Tables 6 and 7 – empty cells in Tables regard to non-investigated associations? If so, please state that in Table caption

·         Lines 558 and 581 – Table references (again, probably due to wrapping)

·         Lines 599-600 – I would suggest some other term instead cross-reactivity (more suitable for immunological methods)

·         Line 631 – Table_S4?

·         Line 640 – we provided reference values

·         Line 641 – we assessed the association

·         Discussion: I would suggest one notion regarding countries for which there are no data (if any) and possible predictions what there could be expected, as risk assessment based on study analyses

·         Discussion: Also, (linked to previous comment) I would suggest one short  paragraph in which authors will emphasize more prediction value of conducted analyses, on infection with TBPs in future  

·         Folder __MACOSX is empty…?

·         Table_S3 – I would just suggest to format the cells to be in one row where needed (for example: Anaplasma centrale Ethiopia Amblyomma lepidum (Teshale et al., 2015))

·         The last reference in the text is 101, and in reference list is 182. I assume that the rest of the references regard to those in supplementary material? If so, please mark them appropriately there.

Reviewer 2 Report

Manuscript titled: Distribution and prevalence of Anaplasmataceae, Rickettsaceae and Coxiellaceae in African ticks: a systematic review and meta-analysis submitted by Carlo Andrea Cossu, Nicola E. Collins, Marinda C. Oosthuizen, Maria Luisa Menandro, Raksha Vasantrai Bhoora, Ilse Vorster, Rudi Cassini, Hein Stoltsz, Melvyn Quan and Henriette Van Heerden provides thorough and comprehensive analysis of prevalence of tick-borne pathogens in Africa. This type of systematic review is significant and provides crucial knowledge on past and current research on tick-borne pathogens in African continent where, as Authors underline, there is a great variation both in terms of research, its clarity and quality. Submitted Manuscript is well prepared and written but unfortunately some flaws occurred.

In Abstract:

L14: Should be “tick-borne pathogens” instead of diseases; or …which are caused by……

L28: replace abbreviation CLEs with full name

Introduction:

L76: a possibility – unnecessary “-“

L92: ….According to  Duron et al.[27]

Materials and Methods:

L133: …provided by Migliavaca et al.[39]

L196-240: clear and visible bullets for each of the described component – its only editorial issue but will help to navigate in text

L231: …according to Egger et al.[43]

L234: Italics in p<0.05

Results:

L323 and other: improve titles and captions of Tables and Figures according to Journal requirements – Check for unnecessary Italics

L403: most occurrences should be replaced with e.g. “highest prevalence”

L450, L459 and other: unify abbreviation of heterogeneity in the whole manuscript

L508: Egger’s

L558, L581 – these sentences require Table number in brackets

L614: this sentence need a clarification. Definitely ticks collected from animals can be partially engorged, but what about ticks collected from vegetation?

L630: Table number is missing in citation

References: Please trace references and check Italics in names of species.

To sum up my opinion, I want to underline the importance of this systematic review in field of tick-borne pathogens in African continent and support the Authors conclusion on usefulness results of this analysis for future epidemiological studies. I believe that after improvement of mentioned flaws, submitted Manuscript should be considered for publication in Microogranisms.
